# Label-Free Microfluidic Impedance Cytometry for Acrosome Integrity Assessment of Boar Spermatozoa

**DOI:** 10.3390/bios12090679

**Published:** 2022-08-25

**Authors:** Stella A. Kruit, Douwe S. de Bruijn, Marleen L. W. J. Broekhuijse, Wouter Olthuis, Loes I. Segerink

**Affiliations:** 1BIOS Lab on a Chip Group, MESA+ & TechMed Institutes, Max Planck Center for Complex Fluid Dynamics, University of Twente, 7500 AE Enschede, The Netherlands; 2CRV Holding BV, 6843 NW Arnhem, The Netherlands; 3Topigs Norsvin Research Center BV, 6641 SZ Beuningen, The Netherlands

**Keywords:** microfluidic impedance cytometry, single cell analysis, acrosome integrity, spermatozoa

## Abstract

Microfluidics and lab-on-chip technologies have been used in a wide range of biomedical applications. They are known as versatile, rapid, and low-cost alternatives for expensive equipment and time-intensive processing. The veterinary industry and human fertility clinics could greatly benefit from label-free and standardized methods for semen analysis. We developed a tool to determine the acrosome integrity of spermatozoa using microfluidic impedance cytometry. Spermatozoa from boars were treated with the calcium ionophore A23187 to induce acrosome reaction. The magnitude, phase and opacity of individual treated and non-treated (control) spermatozoa were analyzed and compared to conventional staining for acrosome integrity. The results show that the opacity at 19 MHz over 0.5 MHz is associated with acrosome integrity with a cut-off threshold at 0.86 (sensitivity 98%, specificity 97%). In short, we have demonstrated that acrosome integrity can be determined using opacity, illustrating that microfluidic impedance cytometers have the potential to become a versatile and efficient alternative in semen analysis and for fertility treatments in the veterinary industry and human fertility clinics.

## 1. Introduction

Artificial insemination plays a major role in veterinary industry to improve reproductive efficiency [1]. Next to closely monitoring and timing the fertilization of females, the sperm quality is of importance and the identification of specific sperm defects is critical for male fertility assessments [2]. In veterinary industry and in human fertility clinics, conventional screening methods include concentration, morphology and motility assessment [3,4,5] viability assays [3,6], acrosome integrity assays [7], and DNA fragmentation assays [8], followed by clean-up methods such as swim-up and density gradient centrifugation [9]. However, the techniques often lack standardization and need expensive equipment, trained and experienced clinicians, and time-intensive processing. Additionally, the viability of spermatozoa and acrosome integrity can be assessed by flow cytometry in combination with staining, the latter being cytotoxic [10]. The results of these assays are diagnostic, and the assessed semen is not suitable anymore in intrauterine insemination (IUI), in vitro fertilization (IVF) and intra cytoplasmic sperm injection (ICSI). Generally, male fertility predictive factors, heavily relying on intensive processing, could greatly benefit from label-free methods to reduce costs, time, and effort. The cells are not harmed during the characterization and could be used for fertilization afterwards.

Fortunately, microfluidic platforms have emerged as versatile, rapid, and low-cost alternatives in a wide range of biomedical applications. The ability to accurately manipulate small amounts of fluids and the high degree of controlling the channel dimensions, have created miniaturized and standardized lab-on-chips [11]. Microfluidics has become a fast-emerging field in the assessment, selection, and sorting of spermatozoa as well [12]. It has been used to analyze the swimming behavior of sperm, such as their motion [13,14,15], rheotaxis [16], and chemotaxis [17,18,19], and other characteristics such as concentration [20,21] and morphology [22]. Even acrosome reaction and DNA integrity were assessed, though still by dye exclusion [23]. Furthermore, sexing of the cells is explored using microfluidic devices [24,25] since the first implementation of X- or Y-sperm enrichment with flow cytometry [26] and free-flow electrophoresis [27] in the 1980s.

Additionally, microfluidic impedance cytometry has made many advances in the label-free characterization of various cell characteristics [28,29,30,31,32,33]. From the real and imaginary signals, features such as magnitude, phase and transit-times are extracted and used to find and identify phenotypic heterogeneity and biophysical characteristics [34]. Depending on the cell type and its surrounding medium, the size of the membrane (2 to 10 MHz), cytoplasm or organelles (10 to 30 MHz) can be probed due to the characteristic frequency dependent dispersions per subcellular region [35]. At low frequencies, the cells are impenetrable by the electric field, while at higher frequencies the polarization of the cell membrane is minimized and cytoplasmic conductivity and permittivity provide the impedance information. Indeed, studies have shown that the impedance magnitude and phase provides valuable information about various cell types, such as their viability [36,37,38,39,40] and intracellular components [41,42].

Spermatozoa are reproductive cells that contain a unique organelle, the acrosome, that is not found in other type of cells. The acrosome is a crucial organelle to facilitate the penetration of the zona pellucida to reach the ovum. This process is known as the acrosome reaction [43]. During the reaction, the outer acrosomal membrane fuses with the plasma membrane leading to the release of acrosomal enzymes and loss of the integrity of the acrosomal cap. If fertilization does not take place, the spermatozoon will die [44]. Tang et al. [42] show in simulations an increase of electrical impedance opacity (magnitude at high frequency over low frequency [45]) for increasing numbers of intracellular components in microalgal cells. With increasing intracellular components, the impedance at low frequencies (0.5 MHz) is hardly affected since the current cannot pass through the cell. At higher frequencies (6 MHz), the polarized membrane allows the current to pass through for intracellular property probing. Vice versa, we hypothesized that the loss of a cellular component, such as the acrosome, would therefore result in an opacity that is lower for cells with a reacted acrosome than with an intact acrosome.

In this study, we applied microfluidic impedance cytometry as a label-free assessment of the acrosome integrity of spermatozoa. The acrosome covers 40 to 70% [46] of the anterior part of the sperms head and is conventionally assessed by labor-intensive methods, such as dye exclusion and high magnification microscopy [10]. Whereas dye exclusion takes specific chemicals, incubation time, imaging and processing, microfluidic impedance cytometry has shown to be a harmless, label-free and a fast approach to characterize various cell types [34]. Though shape characterization of spermatozoa by impedance cytometry is already reported in literature [22], to our knowledge, acrosome integrity assessment by impedance cytometry has not been described before. With the prospect of measuring viability, as was done for other cell types in literature, this microfluidic platform has great potential to become a simple diagnostic tool for fertility assessment.

## 2. Materials and Methods

### 2.1. Sample Preparation

Fresh boar semen samples (breed: Tempo (Topigs Norsvin breeding line), AIM the Netherlands, Vught, The Netherlands)) were obtained at a concentration of 20 × 10^6^ cells·mL^−1^ and stored at 17 °C. The control samples were diluted in 1x phosphate-buffered saline (PBS, Sigma-Aldrich, Buchs, Switzerland) to a concentration of 2 × 10^6^ cells·mL^−1^. The sample was treated with calcium ionophore A23187 (Sigma-Aldrich, Buchs, Switzerland) to induce the acrosome reaction. Briefly, 300 µL of fresh boar sample was centrifuged and the pellet was resuspended in 300 µL Solusem Bio+ sperm diluent (AIM worldwide, Vught, The Netherlands) spiked with 3 mM Ca^3+^ (Sigma-Aldrich, Buchs, Switzerland). The ionophore was added to a final concentration of 50 µM and the sample was incubated for 30 min at 36 °C. After washing, the sample was resuspended in 1x PBS to a concentration of 2 × 10^6^ cells·mL^−1^. The viability and acrosomal status of the control (untreated freshly diluted spermatozoa) and treated sample was evaluated by propidium iodide (PI, Life Technologies, Eugene, OR, USA) and peanut agglutinin antibodies conjugated to Alexa Fluor 488 (PNA-AF488, Invitrogen, Waltham, MA, USA).

### 2.2. Chip Design & Fabrication

The microfluidic chip consists of a glass chip with coplanar microelectrodes, a polydimethylsiloxane (PDMS, Sylgard 184, Dow Corning, Midland, MI, USA) chip patterned with microfluidic channels, and a custom-made printed circuit board (PCB). Coplanar electrodes have the advantage of easy fabrication, but are less sensitive than facing electrodes due to electric field non-uniformity [47,48]. With a few adjustments the sensitivity of coplanar electrodes can be increased [49,50]. In this design [51], the combination of shielding ground electrodes, constriction channels and differential electrodes was used to improve the sensitivity (Figure 1a,b), as similarly described by [52]. The shielding ground electrodes reduce the stray capacitance, thereby increasing the sensitivity at high frequencies. The constriction channels increase the volume fraction of the spermatozoa between the sensing electrodes, while the large electrodes improve the current density. The constriction channel was confined by the coplanar electrodes, spaced 20 μm apart. The height of the channels as well as the width of the constriction channel was 10 μm, creating two sensing zones of 10 × 10 × 20 μm^3^. The fabrication method was adapted from [53] using standard lithography processes.

### 2.3. Chip Preparation & Operation

Before each measurement, a new PDMS chip was bonded using O_2_ cleaning (Harrick PDC-001, Pleasantville, NY, USA) and a custom-made alignment tool. The channels were coated with a poly(L-lysine)-grafted poly(ethylene glycol) (PLL-g-PEG, SuSoS, Dübendorf, Switzerland) solution at a concentration of 100 µg·mL^−1^ in de-ionized (DI) water to prevent the spermatozoa from sticking to the channel walls. After rinsing with the 1x PBS buffer solution (conductivity 1.6 S·m^−1^ at room temperature), the control mixed with 5 µm polystyrene beads (Sigma-Aldrich, Buchs, Switzerland) was introduced into the channels. The beads were used as a reference for each measurement to be able to compare the sample sets with each other. The equivalent diameter of boar spermatozoa is approximately 2.2 μm [54] and due to the size and shape difference, the beads were easily distinguishable from the spermatozoa visually (Figure 1b) and electrically (Figure 1c). After the measurement, the channel was rinsed with the buffer for 10 min before the treated sample-bead mixture was introduced.

The fluidic channels were controlled by a neMESYS syringe pump (Cetoni, Korbussen, Germany) at a flow rate of 0.1 µL·min^−1^ and connected to Hamilton syringes (Hamilton, Reno, NC, USA). The microelectrodes were connected to a HF2LI lock-in amplifier (Zurich Instruments, Zurich, Switzerland) equipped with a HF2TA preamplifier (Zurich Instruments, Zurich, Switzerland). A voltage of 2 V_peak-to-peak_ was applied simultaneously at 0.5 and 19 MHz and the real and imaginary signals were recorded and processed with a custom MATLAB script (R2019a, the Mathworks, Natick, MA, USA).

### 2.4. Data Analysis

At least 200 spermatozoa per stained sample were imaged with an EVOS M5000 microscope (ThermoFisher Scientific, Waltham, MA, USA) and the number of live intact, live reacted, dead intact and dead reacted cells were manually counted. The samples were then introduced into the chip as described above to measure the impedance of, again, at least 200 spermatozoa. The custom MATLAB script identifies the upward and downward peaks of the real and imaginary signal, representing a bead or cell passing the sensing zones (Figure 1c). Due to the size difference in beads and spermatozoa, the signals are easy distinguishable, wherein the beads represent the large peaks and the spermatozoa the small peaks. The average magnitude and phase of the event was derived from both peaks and the data were normalized to the beads to make comparison possible between samples. Lastly, from the normalized magnitudes the electrical opacity [45] was calculated. The significance of the difference in means was examined in IBM SPSS statistics (v. 28, IBM SPSS Statistics, Chicago, IL, USA) with the Welch’s test, as the homogeneity of variances assumption was not met.

The data were visualized in scatter plots to discern clusters of the control and treated spermatozoa in magnitude, phase, and opacity. They were then compared to the results of the staining to find a threshold that would correspond to the determined viability and acrosome reaction for all samples. It was speculated that the threshold that would hold up for all samples was likely to be related to the viability and acrosome reaction, since the viability and acrosome reaction was different for each sample. For all thresholds the performance was measured by means of a receiver operating characteristic (ROC) curve. The sensitivity and specificity were determined from the confusion matrix and classified as:$$Sensitivity=True\;Positive\;Rate\;\left(TPR\right)=\frac{True\;positives}{True\;positives+false\;negatives}$$
$$Specificity=True\;negative\;Rate\;\left(TNR\right)=\frac{True\;negatives}{True\;negatives+False\;positives}$$

This curve specifies the sensitivity and specificity between 0 and 1. In case of a perfect classifier, the sensitivity and the specificity would be 1 (and thus 1-specificity 0). As such, the system is 100% sensitive (no false negatives) and 100% specific (no false positives).

## 3. Results & Discussion

### 3.1. Sample Treatment and Staining Results

In total, four semen specimen were taken from different boars. The semen was approved according to routine processing of artificial insemination in The Netherlands. From each specimen a control and treated sample was checked for viability and acrosome integrity by conventional staining and manual counting (Figure 2). The treatment with the calcium ionophore to induce the acrosome reaction had a strong effect on the viability of the treated sample as expected. The controls showed a low acrosome reaction (5–9%), while the treated samples varied from 37 to 71% reacted (Table 1). These results indicate that each specimen reacts differently to the treatment, also known as acrosome responsiveness or acrosome reaction following ionophore challenge (ARIC) [55].

### 3.2. Impedance Cytometry Results

Using the described microfluidic impedance chip, the complex impedance at multiple frequencies was measured and processed to find useful information on the state of individual spermatozoa of the control sample and treated sample. The normalized magnitude window for which the opacity is analyzed was chosen as such that debris and double cell counts were omitted. For debris, a lower threshold of 0.14 was set; this value was chosen based on the appearance of a spread in opacity below this value and on visual inspection during an experiment. The upper threshold of 0.6 was set, since in the histogram after this value no events were present. The scatter plot of normalized magnitude versus opacity showed that there is an overlap between the control and treated sample (Figure 3a). Yet the treated cells, i.e., induced acrosome reacted cells, have a significant lower opacity (M = 0.82, SD = 0.12) than the control (M = 0.96, SD = 0.08); t(3610) = 44.4, *p* < 0.001, d = 1.47.

Moreover, there is a variation in the number of induced acrosome reacted cells in the treated samples (37 to 71% reacted), which could clarify the wide spread in opacity (Figure 3b). The wide spread could also indicate the state of the acrosome from barely reacted to fully reacted to total acrosomal loss. Furthermore, the acrosome covers about 40–70% of the heads area, which could cause a large deviation from cell to cell in the opacity of the spermatozoa in both samples.

For each individual sample’s scatter plot, a threshold was established for which the number of acrosome intact and reacted cells corresponded to the results of the acrosome staining of that sample (Appendix A). Moving the threshold for each sample to fit best to the staining results revealed that an opacity threshold between 0.82 and 0.93 fits well with the percentage of acrosome reacted cells.

To optimize the opacity threshold for all samples simultaneously, a ROC curve was created. For each control and treated sample, the true positives (TP), false negatives (FN), false positives (FP), and true negatives (TN) were determined and summed to find the sensitivity and specificity of thresholds between 0.4 and 1.2 (steps of 0.01) (Figure 4). The area under the curve (AUC) of this classifier was 0.993. Depending on the application, the importance of sensitivity versus specificity should be considered carefully. For example, if the goal is to get a sample consisting of spermatozoa with only intact acrosomes, the sensitivity should be 1.00. In that case, a threshold of 0.93 is the best choice, with a corresponding specificity of 0.66. This means that all spermatozoa having a reacted acrosome will be classified as that, however, with a downside that 34% of acrosome intact spermatozoa would be incorrectly classified as reacted. Adjusting this threshold to 0.82 would sort all acrosome intact spermatozoa, but 24% of the cells containing reacted acrosomes are sorted into this category as well.

Changing the goal of the platform to be a diagnostic tool, i.e., to measure the acrosome integrity as accurately as possible, the threshold might be chosen differently to compromise between sensitivity and specificity. For example, at a threshold of 0.86 the sensitivity is 98%, while the specificity is 97%. This threshold would give the least false negatives and false positives all together. As shown in Figure 4c, at a threshold of 0.86, the classifier determines the percentage of acrosome reacted in each sample well when compared to the staining, especially when considering a human error of 10% for manual staining and counting. The results of this microfluidic impedance cytometer show a good relation between the opacity and the acrosome integrity of fresh samples.

It is worth noting that the mean low frequency normalized magnitude of treated cells is higher than the mean normalized magnitude of control cells. It indicates that the treated cells have a larger volume and have swollen after treatment. The swelling of cells is characteristic for necrosis and therefore it was considered that viability was measured instead of acrosome reaction. Yet, neither the opacity, nor the magnitude or the phase at the measured frequencies could be related to viability as sensitive and specific as opacity did for acrosome integrity (Appendix A).

Using a similar approach to find an optimal threshold for viability, the phase was found to be the most sensitive and specific. Though the classifier has a high AUC (0.977), in absolute numbers it performs not as good as opacity does for acrosome reaction (Appendix A), with the best tradeoff at a threshold of 0.93 (sensitivity: 88%, specificity: 94%). To enhance the contrast between necrotic and viable cells, buffer optimization is proposed. For instance, Ostermann et al. used a low conductivity buffer and showed great enhancement of the discrimination between viable and necrotic lymphoma cells over a wide range of frequencies (0.5–12 MHz) [36]. However, adjusting the conductivity of our buffer did not give a better discrimination for our cell type (Appendix A).

Ideally our microfluidic impedance cytometer would be able to discriminate between live intact, live reacted, dead intact and dead reacted cells to become a thorough diagnostic tool. Arguably, viability and acrosome integrity are not the only valuable fertility parameters. However, studies have shown that viability is related to DNA fragmentation [56]. If the viability of human sperm is ≤50%, the DNA fragmentation is ≥30%, which indicates subfertility. Vice versa, if the viability is ≥75%, the DNA fragmentation is ≤30%. For acrosome integrity, the in vitro relevance is nuanced. Good sperm samples have low spontaneous acrosome reaction, but it is not predictive of pregnancy outcomes. However, the ARIC score and percentage of induced acrosome reacted sperm are associated with fertilization rate [57]. There seems to be a consensus that acrosome integrity or any other individual parameter is not suitable as a stand-alone fertility test, but is useful in explaining male infertility [10,58].

A great advantage of microfluidic devices is that they are adjustable and extendable. As such, our microfluidic impedance cytometer could be expanded with a few adjustments in the peak finding and fitting algorithm to measure concentration and morphology [22] as well. Though the throughput of the device was low in this proof-of-concept, dedicated hardware and software solutions allow an increase in processing speed [34] to make real-time sorting possible. Similar to computer assisted sperm analysis (CASA), the microfluidic impedance cytometer has the potential to measure multiple sperm characteristics simultaneously. In other words, it would be an affordable alternative for otherwise time-consuming assays and selection methods.

## 4. Conclusions

The opacity of spermatozoa measured with our microfluidic impedance cytometer is associated with acrosome integrity assessed by dye exclusion. Microfluidic impedance cytometry has several advantages over dye exclusion, such as biocompatible media (no toxic chemicals such as staining involved), low volumes, and the possibility to add sorting mechanisms to select spermatozoa for fertilization. Our device has the potential to be easily incorporated into standardized quality checks to save the time, costs, and effort of semen analysis. Future research is needed to find and combine other important sperm characteristics to establish a complete fertility assessor and eventually a selector for fertility treatments in the veterinary industry and in human fertility clinics.

## Figures and Tables

**Figure 1 biosensors-12-00679-f001:**
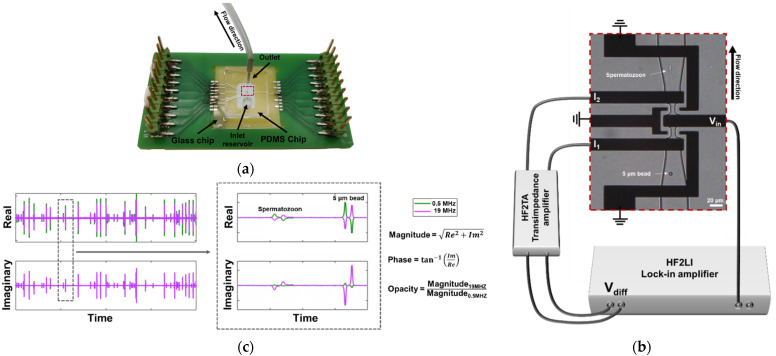
Overview of the chip. (**a**) Microfluidic impedance chip consisting of a PCB, glass chip and PDMS chip. (**b**) Microscopic image of electrodes and channels connected to the impedance spectroscope. The sensing zones are 10 × 10 × 20 μm^3^. An AC voltage is applied to the middle electrode (V_in_), resulting in two currents at the two sensing electrodes (I_1_ and I_2_), which were then measured differentially (V_diff_). (**c**) Example of the real and imaginary signal of a passing bead and spermatozoon, respectively. The data were used to calculate the magnitude, phase and opacity of each event.

**Figure 2 biosensors-12-00679-f002:**
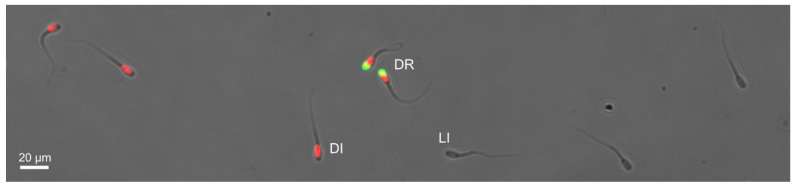
Example image of acrosome staining. No fluorescence indicates live intact spermatozoa (LI). Only bright red fluorescence indicates dead intact spermatozoa (DI). Bright red and green fluorescence indicates dead reacted spermatozoa (DR).

**Figure 3 biosensors-12-00679-f003:**
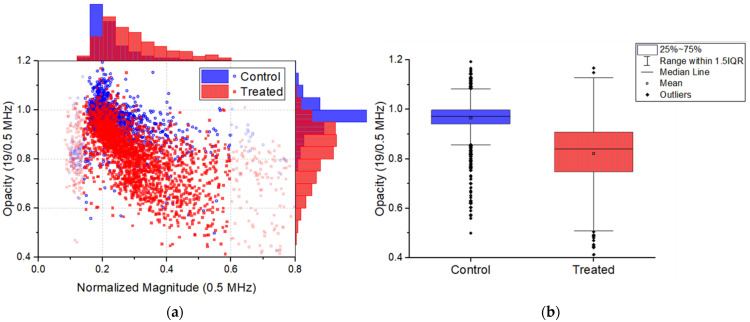
(**a**) Normalized magnitude vs. opacity and the corresponding histograms for all control (**○**) and treated (■) samples. The transparent data points are considered debris or double cells. (**b**) Mean opacity for control and treated sample. Significance level *p* < 0.001, based on the Welch’s *t*-test. See Appendix A for normalized magnitude vs. opacity and normalized magnitude vs. phase for all individual samples.

**Figure 4 biosensors-12-00679-f004:**
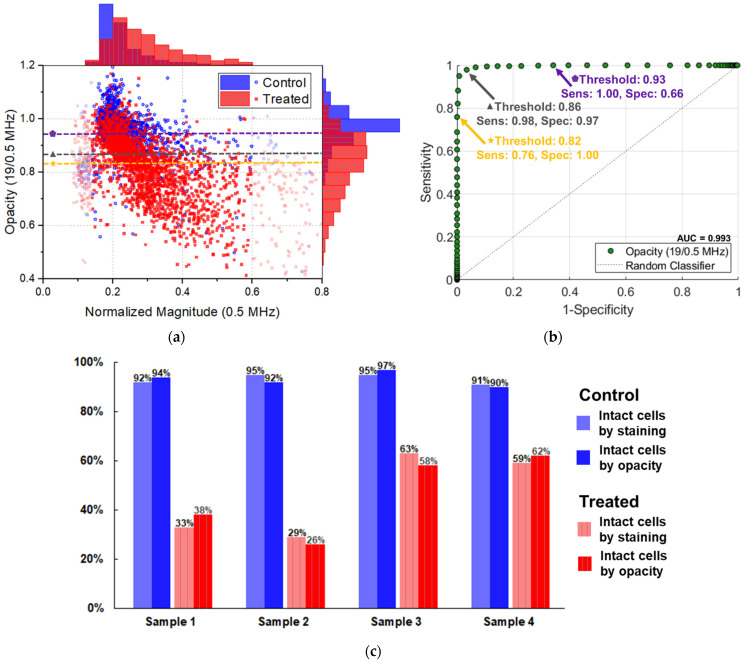
Moving the threshold to find the most specific and sensitive threshold that holds up for all samples. (**a**) Scatterplot with dotted lines representing the thresholds 0.93, 0.86 and 0.82. (**b**) ROC curve of all samples with thresholds between 0.4–1.2, with an area under the curve of 0.993. (**c**) Bar graph to compare of acrosome intact cells by staining and opacity of the control and treated sample at a threshold of 0.86.

**Table 1 biosensors-12-00679-t001:** Live intact, live reacted, dead intact and dead reacted cells of acrosome and PI staining per sample.

Control				
	**1**	**2**	**3**	**4**
Live intact %	76.0	65.7	52.1	52.6
Live reacted %	0.8	0.4	0.7	0.0
Dead intact %	16.0	29.4	42.7	38.3
Dead reacted %	7.2	4.5	5.5	9.1
**Treated**				
	**1**	**2**	**3**	**4**
Live intact %	1.2	7.8	7.0	1.9
Live reacted %	1.6	0.4	0.0	1.2
Dead intact %	32.1	21.2	55.6	57.0
Dead reacted %	65.0	70.6	37.4	39.9

## Data Availability

Data available on request.

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
