# Peer review of "Label-Free Microfluidic Impedance Cytometry for Acrosome Integrity Assessment of Boar Spermatozoa"

_biosensors, 2022, doi:10.3390/bios12090679_

Round 1
Reviewer 1 Report
The authors present a new method for detecting acrosome integrity by impedance spectroscopy in a microfluidic device. The most important thing that was shown in the paper is that this method can detect cells with induced acrosome reaction. The paper sounds good and is well written however several questions must be addressed and minor revision should be performed to improve its impact.
1) authors show that treated cells with induced acrosome reaction are mostly dead. Then motility of such cells is very low. Therefore it seems to be much easier to make standard motility test and separate cells with high motility only. What are the benefits of the method which make it attractive to clinics as it is said in the conclusion.
2) For IVF clinics there are PICSI Dishes for selecting sperm cells with good acrosome. Is it known how cells motility correlates with acrosome integrity? Are there cells with high motility but nonfunctional acrosome?
3) Boar sperm cells and human sperm cells have slightly different morphology. Would your results be applicable for human sperm cells or some adjustment of the method will be required?
4) Signals from the 5 um beads were much higher than from sperm cells. Why you chose 5 um beads? Maybe beads with lower diameter would be preferable to make the signals compareble? What was the signal-to-noise ratio of your measurements? Were the signals, shown on fig 1c, somehow filtered before analysing?
